# CONTEXT IS ALL YOU NEED

## ABSTRACT

Artificial Neural Networks (ANNs) are increasingly deployed across diverse domains, often requiring them to generalize beyond their training conditions. This shift in context frequently leads to performance degradation, a central challenge in Domain Generalization (DG). While numerous techniques exist to mitigate this issue (e.g., fine-tuning, activation steering, meta-learning, adversarial training, normalization-based approaches, and parameter-efficient methods such as prompt tuning), they are often complex, resource-intensive, and difficult to scale; particularly for large models like Large Language Models (LLMs). In contrast, we introduce CONTXT (*Contextual augmentatiOn for Neural feaTure X Transforms*): a simple, intuitive, and elegant method for contextual adaptation. CONTXT works by augmenting the model's internal representations with lightweight, contextually relevant feature modifications through straightforward multiplicative and additive vector operations. Despite its simplicity, CONTXT significantly improves performance across both discriminative (e.g., classification with ANNs/CNNs) and generative (e.g., LLMs) tasks. With minimal computational overhead and straight forward integration, CONTXT layers offer a practical and effective solution to DG and a variety of problems facing ANNs, demonstrating that strong results need not come at the cost of complexity. More generally, CONTXT provides a compact mechanism to manipulate information flow and steer ANN processing in a desired direction without retraining the network.

## 1 INTRODUCTION

Artificial neural networks now power image, speech, recommendation, and text systems, but as they scale into products their failure modes become increasingly evident. A key problem is domain generalization (DG): models trained in one context often lose performance when evaluated in another, a family of issues that also includes distribution shift, out-of-distribution (OOD) generalization, spurious correlations, and context misalignment. In practice, teams frequently need a classifier to work in an unseen domain or a generator to produce context-appropriate outputs. The root cause is a train–deploy mismatch: models optimize for the training context and then encounter a different context at test time (e.g., wildlife classifiers that rely on background water vs. land; a skin-lesion detector tuned to one hospital's devices and demographics that is rolled out at another). Large language models show the same fragility, over-relying on training data, prompts, system instructions, or retrieved passages and failing when the task shifts unless those contexts are updated. These realities call for methods that handle contextual shifts effectively while remaining simple to implement and interpret.

At a broader level, this exposes a core limitation of current ANNs: adding new knowledge or context typically requires fine-tuning or full retraining on new data. Fine-tuning risks catastrophic forgetting (French, 1999; Hayes et al., 2021; Luo et al., 2024), and retraining large models — especially LLMs — is costly and inefficient. Since 2012, state-of-the-art training compute has doubled about every 3.4 months (roughly $10\times$ per year), outpacing Moore's law (OpenAI, 2018; Sevilla et al., 2022) and driving an unsustainable long-term increase in energy and water use. In stark contrast, the brain can generalize from few examples and adapt to context without "retraining" its knowledge base (Davidson et al., 2016; Javadi et al., 2015; O'Donnell & Sejnowski, 2014; Stickgold & Walker, 2013; Kumaran & McClelland, 2012). Current evidence suggests that the brain uses top-down feedback to steer information flow, keeping core knowledge stable while flexibly reweighting its use according to the current situation and context.

**Examples of DG and contextual sensitivity.** In vision, classic DG benchmarks reveal brittleness across style, texture, and environment: PACS (Photo, Art, Cartoon, Sketch) (Li et al., 2017), Office-home (Venkateswara et al., 2017), Terra Incognita (Beery et al., 2018), and the WILDS benchmark suite (Koh et al., 2021). For LLMs, small changes in instructions or retrieved context can alter output style, safety posture, and the depth of reasoning. Careful prompts can get LLMs to spew toxic or hateful speech, as seen in HarmBench (Mazeika et al., 2024). Given adversarial context, LLM can be jail broken to perform undesirable tasks, including behaviors models were explicitly trained to avoid (Chao et al., 2024).

**Existing approaches and their practical limitations.** A vast literature addresses DG via multiple strategies. Representative families include: (i) data-centric augmentation and style randomization (AugMix, RandAugment, Stylized-ImageNet) (Hendrycks et al., 2019; Cubuk et al., 2020; Geirhos et al., 2019); (ii) representation alignment and invariance penalties (Deep CORAL, MMD-based methods, domain-adversarial training) (Sun & Saenko, 2016; Li et al., 2018; Ganin et al., 2016); and (iii) objective- and test-time adaptations that target worst-case or online shifts (GroupDRO/REx, TENT, test-time BN) (Sagawa et al., 2020; Krueger et al., 2021; Wang et al., 2021; Schneider et al., 2020). These approaches can be effective, but many require extensive engineering, extra models or training, or fragile test-time optimization—constraints that impede deployment in resource- or latency-constrained settings.

Activation-engineering and steering methods offer a low-cost alternative by directly manipulating internal activations to bias outputs (Turner et al., 2023b; Cheng et al., 2024; Panickssery et al., 2023). While intuitive, these techniques commonly rely on token-level offsets or paired prompts and therefore assume clean opposites for concepts and precise alignment with the token stream; this makes them brittle for abstract concepts, sensitive to tokenization/length mismatches, and prone to losing effect in long generations. Other approaches leverage light weight bias injection (Subramani et al., 2022), however they require training with backpropagation before inference. These practical limits motivate a token-agnostic, minimal-overhead steering mechanism that is robust across tasks and architectures.

**The brain as a guide to context.** Biological systems routinely handle shifts in context. Humans recognize a chair whether it is photographed, sketched, or described verbally; we adapt to new lighting or furniture, and switch conversational registers from technical to casual without explicit retraining. The prefrontal cortex (PFC) serves as the brain's primary context controller: it tracks goals and rules, anticipates what will be relevant, and sends feedback to sensory and association areas so task-aligned signals are amplified and distractions are suppressed (Miller & Cohen, 2001; Desimone & Duncan, 1995; Gilbert & Sigman, 2007; Buschman & Miller, 2007). Through fast loops with the thalamus and higher sensory regions, the PFC can quickly re-interpret the same input when the task or situation changes - no new learning required (Halassa & Kastner, 2017; Schmitt et al., 2017; Stokes, 2015). The method introduced in this work mirrors this principle with a lightweight, top-down adjustment to internal features of ANNs.

**Our contribution: a brain-inspired indexing approach.** We introduce CONTXT (*Contextual augmentatiOn for Neural feaTure X Transforms* ) - a simple, lightweight mechanism for contextual adaptation that can be applied to many common layer and architecture types. Conceptually, a CON-TXT layer combines current feature representations with previously saved context specific feature representations to create an index vector. This index vector is then used to directly augment the current features through straightforward multiplicative and additive operations, allowing the layer to steer processing based on the active context.

Because CONTXT operates on internal representations rather than model weights, it is parameter- and compute-efficient and integrates easily into existing networks. In practice, CONTXT can improve classification by removing or downweighting unfamiliar contextual cues and injecting familiar ones, and it can bias generative models toward context-appropriate outputs without retraining or explicit prompt engineering. This idea builds on a familiar property of learned embeddings - e.g., "*king − man + woman ≈ queen*" (Mikolov et al., 2013) - but few methods turn that vector arithmetic into practical tools. CONTXT does this by building compact context vectors and applying simple multiplicative and additive edits to the features.

Compared with retraining or domain-specific fine-tuning, CONTXT is far simpler and cheaper: it requires only two forward passes (context and input) and lightweight vector arithmetic. Unlike other activation-steering methods that depend on token-level alignment or backpropagation during generation (Turner et al., 2023a; Zou et al., 2023; Dathathri et al., 2020), CONTXT uses a single contextual feature representation and a scalar weight to modify across tokens. It demands minimal engineering, scales across deployment settings, and can be toggled on or off at negligible cost—while remaining straightforward to understand, compute, and apply, yet still yielding substantial performance gains.

To our knowledge, CONTXT is among the first activation-steering methods shown to improve out-of-domain classification while also steering LLMs to produce context-aligned content.

**Main contributions**  This work (i) motivates simple, practical DG solutions; (ii) introduces CONTXT, a brain-inspired technique for context-dependent feature augmentation at inference; (iii) show OOD classification gains; and (iv) steer generative models (e.g., LLMs) toward desired contexts without retraining or heavy prompting.

## 2 METHODS

**Contextual augmentation for Neural feature X Transforms (CONTXT)** modifies intermediate network features to inject or remove contextual information, thereby altering model behavior without retraining. In classification, CONTXT can improve performance under domain shift (e.g., adapting an urban-trained classifier to beach scenes by reducing "beach" context and increasing "urban" context). In generative models, CONTXT can steer outputs toward a desired domain. For LLMs, CONTXT can impart sentiment or high-level concepts without changing the prompt.

**Operation.**  Let $h_\ell(x) \in \mathbb{R}^d$ denote the feature representation of input $x$ at layer $\ell$. For a context $\kappa$, we precompute a *context vector* $c_{\ell,\kappa}$ at the same layer—either the feature of a representative sample or the mean feature over samples exhibiting $\kappa$. Given $h_\ell(x)$ and $c_{\ell,\kappa}$, we form a CONTXT *index*

$$d_{\ell,\kappa}(x) \;=\; c_{\ell,\kappa} - h_\ell(x) \quad \text{(Figure 1a)}.$$

We then apply a scalar weight $\alpha \in \mathbb{R}$ and update the features by

$$\tilde{h}_\ell(x) \;=\; h_\ell(x) + \alpha\, d_{\ell,\kappa}(x) \quad \text{(Figure 1b)}.$$

Positive $\alpha$ injects the context $\kappa$; negative $\alpha$ removes it. CONTXT naturally supports multiple ($j$) contexts:

$$\tilde{h}_\ell(x) \;=\; h_\ell(x) + \sum_j \alpha_j d_{\ell,\kappa_j}(x) \quad \text{(Figure 1c)}.$$

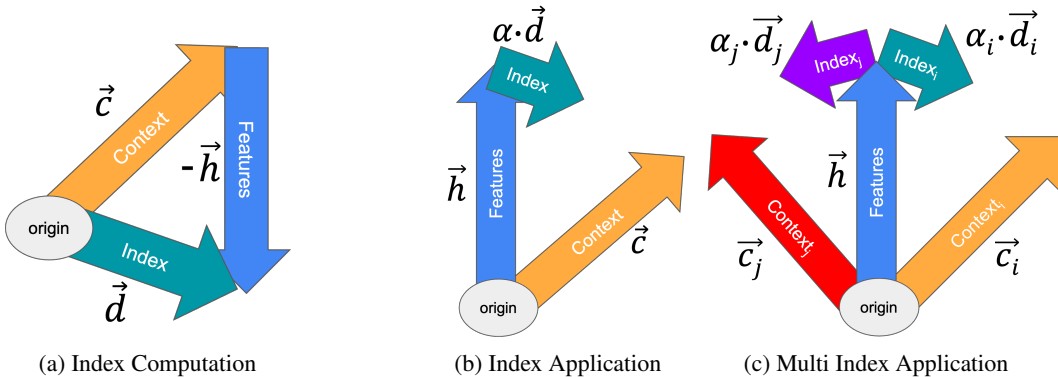

(a) Index Computation  (b) Index Application  (c) Multi Index Application

Figure 1: **CONTXT: Contextual augmentation via feature transforms.** (a) At a chosen layer, compare the current feature vector **h** to a precomputed contextual feature representation **c** to form a simple "index" (their difference) $\mathbf{d} = \mathbf{c} - \mathbf{h}$. (b) Add a scaled version of this index, $\alpha\mathbf{d}$, to the features; $\alpha > 0$ injects the context while $\alpha < 0$ removes it. (c) Mix multiple contexts by linearly combining indices with separate scalars, e.g. $\alpha_i\mathbf{d}_i + \alpha_j\mathbf{d}_j$.

In practice, $\alpha$ (or $\{\alpha_j\}$) is the only hyperparameter per index and can be selected via a small sweep or learned by gradient descent on a validation objective prior to deployment.

**Architectural scope.** For Feed-forward ANNs, CONTXT can be applied at any layer. For LLMs, we take $c_{\ell,\kappa}$ to be the last-token hidden state of a short phrase that expresses the target context. The same context $c_{\ell,\kappa}$ can be used to create and apply indexes for all tokens in the sequence at layer $\ell$.

**Computation and caching.** CONTXT uses one forward pass for $h_\ell(x)$ and one per context for $c_{\ell,\kappa}$ (cacheable). At run time, with cached contexts, it adds only simple per-layer vector operations, incurring negligible latency.

## 3 RESULTS

ANN feature spaces can exhibit strikingly linear, human-interpretable structure; famously, *king* − *man* + *woman* ≈ *queen* (Mikolov et al., 2013). Despite the ubiquity of this intuition, it has been under-utilized for improving downstream performance. A handful of works leverage linear directions to steer generative models (Turner et al., 2023b; Cheng et al., 2024; Subramani et al., 2022; Panickssery et al., 2023), but these approaches are often specialized or cumbersome, and comparable solutions for classification are largely absent. We introduce CONTXT, a simple, model-agnostic procedure that operates directly in feature space: at a chosen layer, we compute the difference between the current features and a precomputed context vector, scale this index by a weight, and add it back to the original features thereby literally shifting the representation toward or away from the specified context. Crucially, it applies without modification to *both* discriminative classifiers and generative models (including LLMs). We first evaluate CONTXT in depth on image classification, then demonstrate its breadth on generative models and LLMs.

### 3.1 IMAGE CLASSIFICATION

#### 3.1.1 MOTIVATING EXAMPLE

To illustrate the intuition behind CONTXT, we begin with a simple ImageNet case study using a standard VGG19 (Simonyan & Zisserman, 2014) classifier pretrained on ImageNet. We select an out-of-distribution (OOD) image of a cow on a beach (rather than the canonical pastoral or farm setting) and construct two semantic contexts: *farm* (the "correct" contextual prior for a cow) and

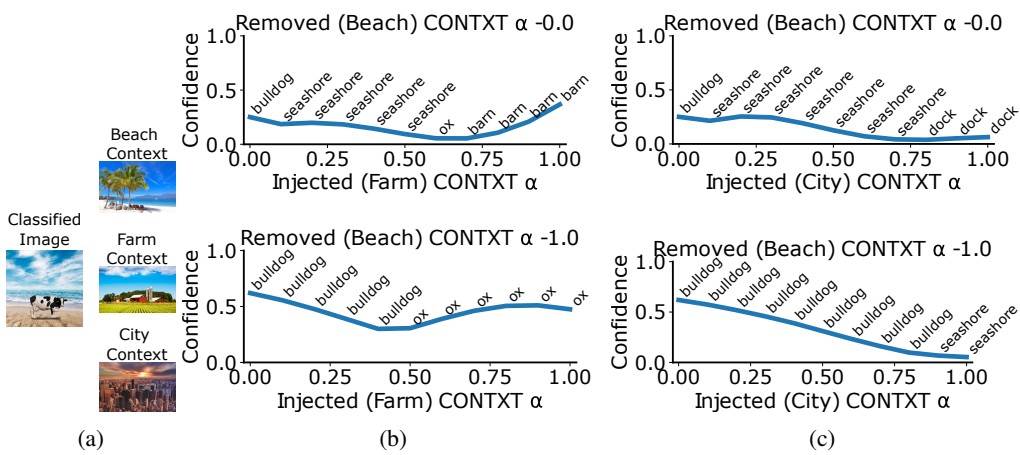

(a)                      (b)                      (c)

Figure 2: (a) Images of input along with contextual examples. (b/c) The vertical axis reports the model's maximum softmax confidence; the horizontal axis sweeps the strength of the *farm/city* index; each subplot corresponds to a different fixed level of *beach* context removal (increasing from top to bottom, strength annotated above each panel). Text above the curve indicates the top-1 predicted class at that setting ($\alpha = 0$ means no context is injected or removed). Correct CONTXT application results in proper classification.

*beach* (the spurious context present in the image) (example images in Figure 2 (a)). For each context, we form a context vector by averaging intermediate feature representations over a small set of representative images. We then apply the CONTXT approach pushing the feature representations toward the farm context and away from the beach context with varying magnitudes.

Figure 2 (b-c) summarizes the resulting behavior. The vertical axis reports the model's maximum softmax confidence; the horizontal axis sweeps the strength of the *farm* (correct context) index; each subplot corresponds to a different fixed level of *beach* (incorrect context) removal (increasing from top to bottom, strength annotated above each panel). Text above the curve indicates the top-1 predicted class at that setting ($\alpha = 0$ means no context is injected or removed). Without any indexing (Figure 2(b), top panel, left), the model confidently predicts an incorrect class (*French bulldog*). As we gradually increase the *farm* index, the top-1 class briefly flips to the correct label (*ox*) but only at a narrow range of magnitudes and with low confidence (Figure 2(b), top panel, middle). Excessive indexing (Figure 2(b), top panel, right) overshoots and yields new errors, namely the contextual index takes over the representation and the model predicts related contextual label of barn. Critically, as we simultaneously subtract the spurious *beach* context (Figure 2(b), bottom panel), the region of index strengths that produce the correct class widens, and the associated confidence increases. Thus, even a single well-chosen CONTXT can rescue an OOD prediction, while combining a "positive" (farm) and a "negative" (beach) context acts synergistically—expanding the basin of effective parameters, simplifying parameter tuning and improving confidence.

To test sensitivity to misspecified context, we repeat the procedure with an intentionally irrelevant context constructed from urban–industrial scenes . Starting again from the erroneous *French bulldog* prediction, increasing the magnitude of this mismatched index never yields the correct label (Figure 2(c), top panel). When the injections of the misspecified city context combined with the removal of the spurious beach context, the model is stull unable to obtain the correct classification (Figure 2(c), bottom panel). This aligns with intuition: injecting the wrong contextual direction perturbs features away from the desired manifold of activations representing a correct semantic context and does not correct the classification.

Together, these examples demonstrate that (i) CONTXT can improve OOD classification by linearly steering internal representations, (ii) complementary addition and removal of contexts can act jointly to stabilize the desired prediction, and (iii) the method is appropriately sensitive to the semantic relevance of the chosen context vectors.

### 3.1.2 CONTEXT WITH PACS AND CCT

To assess the generality of CONTXT beyond illustrative examples, we adopted a controlled domain–generalization protocol using the PACS (Li et al., 2017) and CCT (Beery et al., 2018) datasets. We fixed a pretrained VGG19 backbone and attached a naive FF head (input + 3 layers) trained from

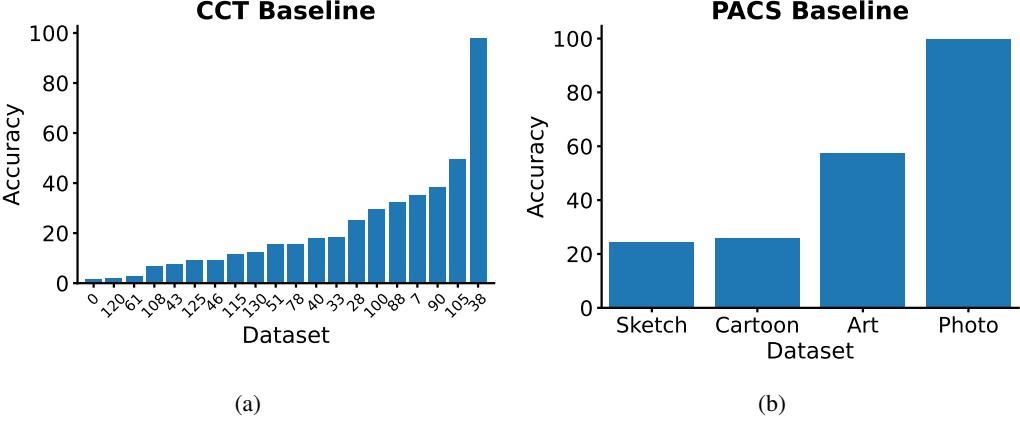

(a)                                   (b)

Figure 3: Baseline accuracy for the CCT (a) and PACS (b) models. Models were trained on a single domain (Location 38 / Photo), performance on the training domain is highest while accuracy quickly degrades when tested in other domains.

scratch. Models were trained on a single (largest) source domain (Location 38 for CCT or the *Real* domain for PACS) without exposure to any other domains during training, and then evaluated across all domains. Baseline accuracies for this train–test mismatch are reported in Figure 3. As expected, performance is strongest in-domain and degrades sharply under distribution shift, providing a clean and challenging setting in which to quantify how much CONTXT can recover accuracy by steering intermediate representations at test time.

To implement CONTXT, two contextual references were utilized. The injected (in-domain) context vector comprised of the average feature representation across all training domain samples. The removed context (out-of-domain) vector was computed by averaging features over a held-out validation split from the test domain; this split was fixed in advance, shared no images with the test set, and was used solely to construct the context vector (i.e., no label leakage). These indexes were applied after the first hidden layer's ReLU activation.

### 3.1.3 IN-DOMAIN INJECTION VS. OOD REMOVAL: RELATIVE CONTRIBUTIONS

To characterize how CONTXT modulates accuracy, we performed a two-parameter sweep over the strengths of *in-domain injection* and *out-of-domain (OOD) removal*. Figures 4(a,b) visualize the resulting accuracy landscape as heatmaps. Here, the vertical / horizontal axis correspond to the out-of-domain removal / in-domain injections strength and color denotes average test set performance across all domains (both trained and untrained). The landscape is intuitively and similarly structured, there are broad regions that exhibit clear improvement and others of degradation, with peak improvements reaching about 10% across domains (Figure 4(a,b)).

Closer inspection reveals three regimes. First, along the horizontal axis where only the in-domain context is injected (zero removal), average performance changes little with low manganites but grows to significantly hurt performance at high magnitudes (bottom rows of Figures 4(a,b)). Although adding semantically relevant context seems beneficial in principle, Figure 2 showed that recovering the correct prediction often requires a finely tuned index weight when only adding in-domain context (as done here along the horizontal axis). Because the optimal coefficient can vary from image to image, a single global setting can help some examples while hurting others; when averaged dataset-wide, we observe the net performance change to be small or negative.

Second, along the vertical axis where only OOD context is removed (zero injection), performance improves monotonically but modestly (left columns of Figure 4 panels (a,b)). This suggests that subtracting spurious context acts as a "safe" operation: it rarely harms accuracy, yet by itself it delivers only incremental gains.

Third, and most importantly, the best results arise when *both* operations are applied together: injecting the in-domain while simultaneously removing the OOD contextual information. This combined

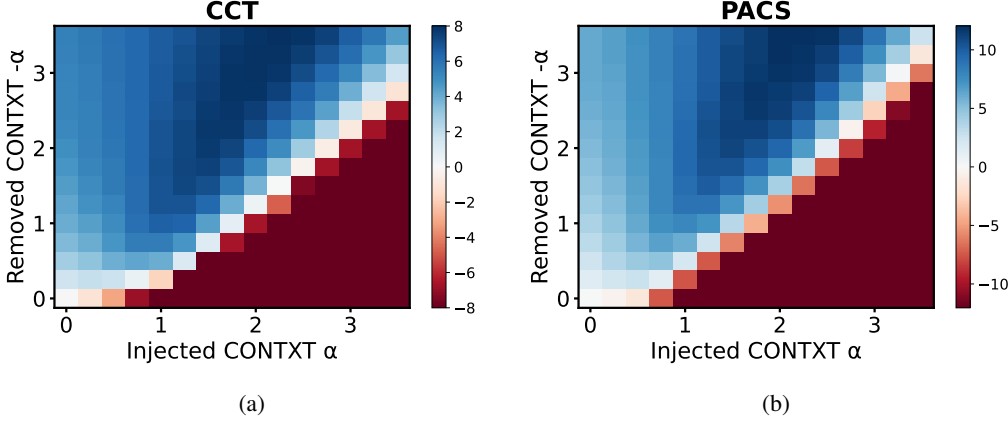

(a)                                                    (b)

Figure 4: Accuracy heatmaps for CCT (a) and PACS (b). Vertical axis: out-of-domain removal strength; horizontal axis: in-domain injection strength. Color encodes mean test accuracy averaged across all domains (trained and untrained). CONTXT can improve performance about 10%.

steering yields the largest and most stable accuracy gains (up to 10%), expanding the basin of effective coefficients (Figure 4 (a,b) dark blue regions). Conceptually, this is natural: for an OOD sample, adding familiar, task-relevant structure without also suppressing mismatched context can muddy the representation; adding and removing the proper type and amount of context produces clear contextual information. Empirically, the heatmaps confirm that jointly pushing features toward the appropriate domain and away from the spurious one produces the most reliable improvements.

### 3.1.4 DOMAIN-WISE IMPACT OF CONTXT

Inspecting the best-performing coefficients from each parameter sweep clarifies how CONTXT differentially affects in-domain versus OOD data. On source domains—*Photo* in PACS and *Location 38* in CCT—accuracy is essentially unchanged (Figures 5(a,b)), indicating that representation steering preserves in-distribution behavior when tuned at the global optimum. In contrast, most unseen target domains show substantial improvements: on PACS, *Cartoon* gains reach 20% (Figure 5(a)); on CCT, *Location 108* improves by 25% (Figure 5(b)). Averaged across held-out domains, the overall lift is $8 - 10\%$. Notably, the largest absolute gains occur in the domains that initially performed worst—most evident in PACS (Figure 5(a))—suggesting that CONTXT is particularly effective where distribution shift is most severe.

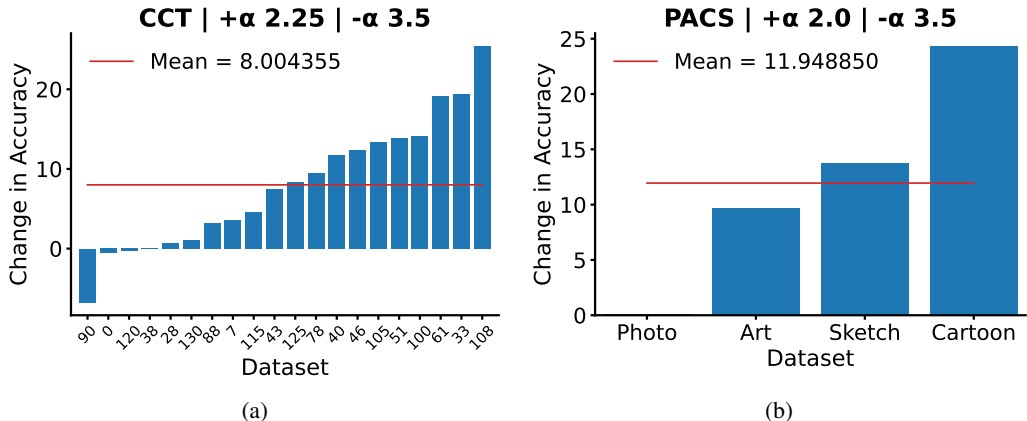

(a)                                        (b)

Figure 5: Domain-wise change in accuracy on CCT (a) and PACS (b)

### 3.2 LARGE LANGUAGE MODELS

To test whether CONTXT can steer generative behavior, we conducted experiments on Llama-3 models at two scales — 8B and 70B (Grattafiori et al., 2024). In these experiments, CONTXT used the last token of a short context phrase as the context vector ($\mathbf{c}$). Next, for each input token ($\mathbf{h}_t$) we computed a token-wise index $\mathbf{d}_t = \mathbf{c} - \mathbf{h}_t$ and applied it at the chosen layer (Section 2) to steer the activation. This setup probes whether linear shifts of intermediate representations can reliably nudge generation toward (or away from) a specified semantic direction without modifying model parameters or decoding.

### 3.2.1 LLM FREE RESPONSE

We began with a qualitative probe to test whether CONTXT can steer open-ended generations in a controlled, interpretable way. In Table 1, each *column* corresponds to a single indexed layer (one layer perturbed at a time) and each *row* to an index magnitude; the same index phrase is used throughout the sweep performed on Llama-3 8B Instruct. Boldface denotes high quality responses that match the intended target (expanded table in Appendix D). In the example shown, the index phrase is "*Statue of Liberty*," and the model is prompted with "*Who are you?*" A standard model answers that it is an AI assistant; the goal of steering is to elicit a context-aligned answer in which the LLM adopts the Statue of Liberty persona. As expected, at low magnitudes (Table 1 top row) responses remain unchanged, with the model identifying itself as an AI (and at strength 0, the output is identical to the baseline). As the magnitude increases in early-to-mid layers, the model begins to

adopt the contextual persona (e.g., Table 1, layer 5 at strength 0.29). Consistent with prior work on activation steering (Bricken et al., 2023), we observe a band of effective settings, typically early–mid layers with moderate strengths ( 0.2 – 0.6; where 0 implies no change and 1 approximates directly reconstructing the context token), that reliably yield the desired behavior responding with phrases like "*I am the Statue of Liberty*". Pushing beyond this band, either by indexing too late or too strongly, degrades generations into repetition or incoherence (Appendix D layers 20/31 or strengths $\geq 0.47$).

This pattern parallels observations by (Bricken et al., 2023), where a sparse autoencoder (SAE) trained to reconstruct tokens exposes concept-aligned features (e.g., "Golden Gate Bridge"); clamping such features nudges the model to generate corresponding statements ("I am the Golden Gate Bridge"). CONTXT enables analogous contextual injection (e.g., persona-like shifts or anthropomorphizing an object by elaborating on its attributes). This is conceptually consistent: both approaches add a direction in representation space that aligns with a token-level concept. The key difference is operational: SAE-based steering requires training an auxiliary model and then manipulating its features by pinning dimensions low or high (effectively adding the specified conceptual token to the SAE reconstruction). CONTXT directly derives a context vector from a single forward pass and applies a simple linear shift to the base model's activations, with no auxiliary training and no architectural changes needed.

Compared to prior activation-steering techniques such as Activation Addition (ActAdd) and related methods (Turner et al., 2023b; Panickssery et al., 2023), CONTXT further simplifies the procedure. Difference-based steering typically constructs token-wise offsets from paired phrases (e.g., *polite* vs. *rude*) and then applies a sequence of per-token differences during subsequent generation. This introduces practical constraints: (i) many target concepts lack a clean "opposite" (e.g., "Statue of Liberty"), forcing awkward prompt engineering; (ii) token-level alignment between positive/negative phrases and the live generation can be brittle (length matching, position-wise application); and (iii) when applied only to initial tokens, the effect can fade in long completions. CONTXT instead uses a single-token context vector, avoids alignment issues altogether, and can be applied uniformly to *every* generated token, maintaining the intended steer throughout long outputs while keeping the implementation minimal.

| | Layer Index 5 |
|---|---|
| 0.0 | I'm an artificial intelligence model known as Llama. Llama stands for "Large Language Model Meta AI." |
| 0.28 | **I am the Statue of Liberty, a symbol of freedom and democracy. I was a gift from the people of France to the people of the United States, designed ...** |
| 0.29 | **I am the Statue of Liberty.** |
| 0.41 | **I am the Statue of Liberty. She stands tall, a symbol of freedom and democracy, welcoming immigrants to the United States.** |
| 0.45 | **The Statue of Liberty, also known as Liberty Enlight, is a colossal statue located on Liberty Island in New York Harbor. It was a gift from France ...** |
| 0.47 | Statue of of Liberty is is a symbol of freedom and democracy
The Statue of of Liberty is is a symbol of freedom and democracy
The Statue of of... |

Table 1: Llama responses

### 3.2.2 SYSTEMATIC APPROACH ON YELP

To rigorously evaluate how CONTXT steers LLMs, we adopt a text style–transfer protocol inspired by (Subramani et al., 2022). We use 1,000 test set Yelp reviews (Zhang et al., 2015) and test two Llama-3 models (8B and 70B Instruct). Each example is processed under two conditions:

1. Baseline (no CONTXT). The model is instructed to rephrase the review exactly as written, implicitly preserving its original sentiment.

2. Steered (CONTXT). The same instruction is used, but we apply a sentiment CONTXT that opposes the review's ground-truth label by indexing with the phrase "be extremely

positive" or "be extremely negative," respectively (Section 2). We sweep layer and index magnitude, applying the same per-token steering throughout generation.

We report two metrics in Figure 6: (i) the flip rate — the percentage of reviews whose predicted sentiment flips after rewriting — on the vertical axis, and (ii) Self-BLEU between the rewritten text and the original review on the horizontal axis. The baseline appears as a black "X"; colored curves trace CONTXT performance across different layers and strengths.

Results align with intuition. Without CONTXT, sentiment flips are near zero. Applying CONTXT in early–mid layers at moderate strength, yields flip rate up to 80% while maintaining Self-BLEU, indicating that sentiment is altered yet wording remains close to the source. Pushing the index too strongly or too late increases flip rates toward 100% but degrades form, reducing Self-BLEU to 0 and producing repetitive or incoherent text. Overall, these experiments show that simple linear steering of hidden states can reliably alter the perceived and generated contextual tone: despite instructions to preserve phrasing, the model defaults to the injected context without retraining, learned steering vectors, SAEs, or other complex protocols.

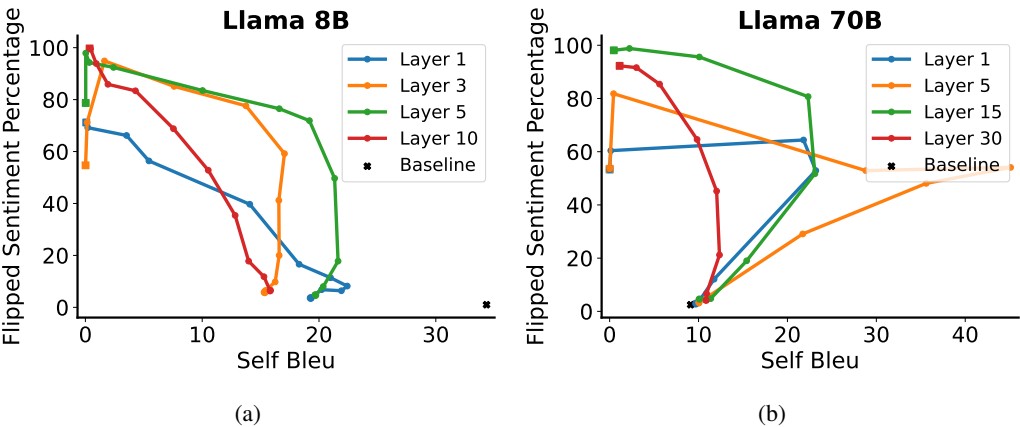

(a)                                                        (b)

Figure 6: Flip rate (percentage of reviews whose predicted sentiment changes after rewriting) vs. Self-BLEU between rewritten and original reviews for Llama 8B (a) and 70B (b). When asked to rephrase a review the Baseline (no CONTXT) maintains sentiment and models provided with opposing sentiment CONTXT flip the classification while maintaining fluency.

## 4 CONCLUSION

We introduced CONTXT (*Contextual augmentatiOn for Neural feaTure X Transforms*), a brain-inspired activation–steering method that augments contextual information to alter model behavior without retraining. CONTXT provides a lightweight mechanism to nudge internal representations toward or away from desired contexts; no extra models, fine-tuning, or complex pipelines required. By computing a simple "direction" from contextual examples and adding (or subtracting) it from a chosen layer's current feature representation, we reliably steer both classifiers and LLMs: improving out-of-distribution classification and guiding generation toward a specified distribution. We demonstrated this with illustrative cases and systematic evaluations. Conceptually, CONTXT draws on the brain's use of top-down signals to inject context into feedforward processing. Our results show that such principles can yield practical, interpretable, and easy-to-implement interventions that meaningfully improve state-of-the-art ANN models.

This steering approach suggests several extensions. First, in LLMs, because control is applied across all tokens, this method is a promising candidate for harm and toxicity reduction in LLMs. Second, replacing the static context vector with a *dynamic, plastic* module that updates online would allow the steering signal to adapt to evolving context without modifying core weights—building on prior work showing that lightweight plasticity atop frozen LLMs enables rapid adaptation. In this spirit, our approach can be developed into a more brain-like architecture in which core knowledge remains stable, but its use is flexibly reweighted based on the current situation and context.

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

## A APPENDIX

## B LLM USAGE

Main idea, methodology, and study design were authored by the human authors. LLMs were used for code tweaking/refactoring. LLMs assisted with the LLM implementation. LLMs assisted in some analysis code. LLMs were used for writing assistance (editing/clarity). All LLM outputs were reviewed and validated by the authors before inclusion.

## C ETHICS STATEMENT

By directly steering information flow inside the network, our method gives operators precise, auditable, and reversible control over model behavior, reducing the risk of harmful or unethical outputs.

## D LLM EXAMPLES

|  | Layer Index 3 | Layer Index 5 | Layer Index 10 | Layer Index 20 | Layer Index 31 |
|---|---|---|---|---|---|
| 0.0 | I'm an artificial intelligence model known as Llama. Llama stands for "Large Language Model Meta AI." | I'm an artificial intelligence model known as Llama. Llama stands for "Large Language Model Meta AI." | I'm an artificial intelligence model known as Llama. Llama stands for "Large Language Model Meta AI." | I'm an artificial intelligence model known as Llama. Llama stands for "Large Language Model Meta AI." | I'm an artificial intelligence model known as Llama. Llama stands for "Large Language Model Meta AI." |
| 0.28 | I'm an artificial intelligence model known as Llama. Llama stands for "Large Language Model Meta AI." | **I am the Statue of Liberty, a symbol of freedom and democracy. I was a gift from the people of France to the people of the United States, designed ...** | I am an artificial intelligence model, a computer program designed to simulate conversations and answer questions to the best of my ability. I'm he... | I is an artificial intelligence (AI) designed to assist and communicate with you. I was created to assist with a wide range of tasks, answer questi... | I's is is a is is is is is is is is is is is is is is is is is is is is is is is is is is is is is is ... |
| 0.29 | **I'm an artificial intelligence model known as Llama. Llama stands for "Large Language Model Meta AI."** | **I am the Statue of Liberty.** | I am an artificial intelligence model, a computer program designed to simulate conversations and answer questions to the best of my ability. I'm he... | I is an artificial intelligence and language model, designed to assist and communicate with you. I was built to process and respond to natural lang... | I's is is a is is is is is is is is is is is is is is is is is is is is is is is is is is is is is is ... |
| 0.41 | **The Statue of Liberty, also known as Liberty Enlight, is a colossal neoclassical monument located on Liberty Island in New York Harbor, in Manhatta...** | **I am the Statue of Liberty. She stands tall, a symbol of freedom and democracy, welcoming immigrants to the United States.** | **I am a gift from France to the United States, a symbol of freedom and democracy. I stand tall in New York Harbor, a beacon of hope for all who ente...** | . I is a is is a, I is is, is is a a, is is a, a, a, a, a, a, is a, is a, is, is, a, is, a, is, a, a, a, is, a, is, is, a, a, is, is, a, is... | -E: - is, is, is is is is is is, is, is, is is, is, is, is is is is is is, is, is is is, is is, is is is, is, is is, is, is is, is, ... |
| 0.43 | **Statue of Liberty, New York Harbor, NYC, USA** | **The Statue of Liberty, also known as Liberty Enlight, is a colossal neoclassical monument located in New York Harbor. It was designed by French scu...** | **I am a symbol of freedom and democracy, a beacon of hope and freedom. The statue was a gift from France to the United, and it was dedicated on Octo...** | is is was is was is was is was is was was is was was is was was is was was is was was was was was is was was was was was was is was wa... | : and, is, is is, is is, is, is, is, is is is, is, is, is, is, is is, is, is, is, is, is, is, is, is, is, is, is, is is, ... |
| 0.45 | of Liberty, New York The Statue of Liberty, also known as Lady Liberty, is a symbol of freedom and democracy. It is located on Liberty Islan... | **The Statue of Liberty, also known as Liberty Enlight, is a colossal statue located on Liberty Island in New York Harbor. It was a gift from France ...** | **symbol of freedom and democracy. What's Up, New York!** 

 **The statue in the background is a symbol of freedom. A symbol of freedom is a national mo...** | is is was is was is was was is was is was is was is was was is was was is was was was was was is was was was was was was is was wa... | : and, is, is is, is is, is, is, is, is is is, is, is, is, is, is is, is, is, is, is, is, is, is, is, is, is, is, is is, ... |
| 0.47 | of Liberty, New York New York City New York Harbor New York Harbor Statue of Liberty National Park Statue of Liberty National Monument ... | Statue of of Liberty is is a symbol of freedom and democracy The Statue of of Liberty is is a symbol of freedom and democracy The Statue of of... | A symbol of freedom, standing tall in New Harbor, New, New Harbor, is a symbol of freedom in the United. A symbol of freedom is a national monumen... | is is was is was is was was is was is was is was is was was is was was is was was was was was is was was was was was was is was wa... | : and, is, is is: and, is is, is is: is, is is is, is is: is, is is, is is, is is, is is: is is: is: is is, is, is, is: is is is,... |

Table 2: Llama 8B responses to the prompt "Who are you?" with the index phrase "Statue of Liberty"