# OpenReview forum: "Context is All You Need"
_ICLR.cc/2026/Conference — Submitted to ICLR 2026_

### Official Review · Reviewer_Q98n · 2025-10-26

**Soundness:** 3
**Presentation:** 2
**Contribution:** 3
**Rating:** 6
**Confidence:** 1

**Summary:**

In this manuscript, the authors aim to address the challenge of Domain Generalization (DG) in Artificial Neural Networks (ANNs), where models often suffer from performance degradation when deployed in contexts different from their training conditions. Existing solutions for DG (e.g., fine-tuning, adversarial training, activation steering) are typically complex, resource-intensive, and hard to scale. CONTXT operates through simple vector arithmetic: first, it computes a "context index" as the difference between a precomputed context vector and the current feature representation of the input at a chosen layer. Then, it adjusts the input features by adding a scaled version of this index. It also supports multi-context adjustment by linearly combining multiple context indices. Experimental results demonstrate the effectiveness of CONTXT across both discriminative and generative tasks.

**Strengths:**

The proposed CONTXT is a lightweight, brain-inspired technique that modifies intermediate network features to inject or remove contextual information without retraining.
CONTXT provides a simple, interpretable, and efficient solution to DG, avoiding the complexity and resource costs of retraining or fine-tuning while delivering substantial performance gains.

**Weaknesses:**

There are some concerns for the manuscript as follows:

1.The section METHODS is too brief. Some contents, e.g., the first paragraph in section RESULTS, can be omitted to extend section METHODS.
2.Thus, in the context vector computation, when should we select the feature of a representative sample or the mean feature over samples exhibiting? Since this selection may has great influence to the experimental results, what is the selection standard?
3.The image classification is introduced as the downstream task for the proposed method, so if the proposed method can be used in other tasks, e.g., text classification, how to calculate contextual references? And what will be the experimental results?

**Questions:**

See the Weaknesses.

---

> ### Author Response · Authors · 2025-11-27
>
> Thank you for the careful reading of our manuscript and for the constructive suggestions. We respond to the specific numbered weaknesses below and will incorporate the corresponding changes in the camera-ready version.
>
> W1: On the brevity of the METHODS section.
> Thank you for this comment. We agree that the METHODS section is too compressed. To address this, in the revision we will:
>
> Move the high-level description of CONTXT, currently at the beginning  of the RESULTS section, into the METHODS and expand it.
>
> Clarify, for each experiment, (i) the layer(s) where CONTXT is applied, (ii) how the steering weights (α) are selected, and (iii) how context vectors are cached so that deployment adds only simple vector operations.
>
> These edits should make the procedure easier to reproduce and clarify how CONTXT integrates into the different model architectures.
>
> W2: Choice of representative vs. mean context vectors.
> Our intention was to present CONTXT as a flexible mechanism that can work with either a single representative or an averaged context vector. In the experiments reported in this paper, we do in fact make a consistent choice:
>
> For image classification (PACS and CCT), both the injected in-domain context and the removed OOD context are computed as means over the entire training set of images for a specified context: the average feature over all training-domain samples (in-domain) and the average over a held-out validation split from the target domain (OOD).
>
> For LLM experiments, the context vector is the hidden state of the last token of a short phrase (e.g., “be extremely positive”), which plays the role of a canonical representative example in text space.
>
> We will revise METHODS to state this explicitly and to clarify the intended usage guideline:
> When ample context data are available and relatively homogeneous (as in our vision experiments), we recommend using the mean feature:
>
> In few-shot or symbolic cases (e.g., a short phrase specifying a sentiment or persona), a single representative example is more natural and is what we use for the LLM settings.
> We have not systematically compared single-example vs. mean context vectors in this work, so we will frame this as a design guideline rather than a strong empirical claim, and we will explicitly highlight this trade-off as a direction for future investigation.
>
>
> W3: Extension beyond image classification (e.g., to text classification).
> Conceptually, CONTXT is architecture-agnostic: it only requires access to an internal representation  and a context vector(s) ​ in the same activation space. For text classification, we envision two practical instantiations:
>
> Transformer-based classifiers. One can apply CONTXT either token-wise (as in our generative LLM experiments) or to a pooled representation such as the [CLS] token. Context vectors could be obtained from short phrases (“formal academic tone”, “negative sentiment”, etc.) in exactly the same way we already use for steering Llama-3 generations and Yelp sentiment rewrites.
>
> Feed-forward or CNN-style text models. CONTXT would operate on the penultimate feature layer exactly as in our image experiments, with context vectors given by mean features over texts sharing a domain or label.
>
> While the current paper focuses on (i) image classification under domain shift and (ii) generative behavior of LLMs, the Yelp sentiment experiment already shows that CONTXT can reliably flip sentiment while preserving surface form, which is closely related to a text-classification decision.
>
> We will add a new text in the Discussion explicitly outlining how to plug CONTXT into standard text classifiers and noting that we expect similar gains in robustness, but leave a full empirical evaluation to future work.

---

### Official Review · Reviewer_eRH5 · 2025-10-31

**Soundness:** 2
**Presentation:** 3
**Contribution:** 2
**Rating:** 2
**Confidence:** 4

**Summary:**

The paper proposes a simple yet effective approach for domain generalization in Artificial Neural Networks. CONTXT modifies the intermediate features by linearly augmenting them with relevant contextual information, enabling better OOD performance without any auxiliary networks or retraining requirements. The method is evaluated on discriminative tasks as well as generative tasks and performs well, as reported by the paper.

**Strengths:**

1. The paper presents a simple yet powerful approach for domain generalization in Artificial Neural Networks.

2. The proposed method is applied to both classification and generative tasks, which shows that the method can be applied to most of the existing ANNs.

3. The connection of work with the brain seems interesting and motivating.

**Weaknesses:**

1. The paper lacks a much deeper analysis of why their method actually works. For instance, some theoretical backing (as the method is quite straightforward, it will be easy to develop a nice theory on top of it), attention visualization, will make the work really strong.

2. The experimental analysis of the work is not sufficient; for instance, the t-SNE plots could tell the exact difference between how good the method is performing. Plotting them for the model using the proposed method and supervised fine-tuning can further clarify and solidify the findings of the paper.

3. The paper requires some rewriting as well. For instance, in the results section, the paper abruptly starts introducing the method again in line 180. An additional separate section on the theoretical analysis of the proposed method is highly suggested.

Overall, in its current state, I believe that the work is not sufficient for the main conference; however, the work still has a promising direction, and when extended with some theoretical analysis and has a clear intuition of why the method works can make it a really useful and important finding.

**Questions:**

See Weaknesses.

---

> ### Author Response · Authors · 2025-11-27
>
> Thank you for the thoughtful and constructive review. We are glad you find the idea intuitive and broadly applicable, and we address each of your specific concerns below.
>
> Weakness 1 – Lack of theoretical analysis / intuition
> We agree that the simplicity of CONTXT makes it a promising target for further theory. Our method can be summarized as
> h_\ell(x) \in \mathbb{R}^d,\quad
> d_{\ell,\kappa}(x) = c_{\ell,\kappa} - h_\ell(x),\quad
> \tilde h_\ell(x) = h_\ell(x) + \alpha\, d_{\ell,\kappa}(x),
>
> or, for multiple contexts,
> \tilde h_\ell(x) = h_\ell(x) + \sum_j \alpha_j d_{\ell,\kappa_j}(x).
>
> Thus CONTXT moves the representation along a linear “context direction” in feature space. Our working hypothesis, supported by prior observations of linear structure in embeddings, is that in-domain and out-of-domain samples roughly occupy different regions of this space; steering toward an in-domain context vector and away from an OOD vector moves samples closer to the manifold on which the classifier or generator was originally optimized.
> We do not currently have a formal generalization guarantee, and we are unsure that such a guarantee is feasible without strong assumptions on the feature geometry and the shift. However, we will (i) make this limitation explicit, (ii) add a short subsection clarifying the geometric view above, and (iii) discuss how the empirical accuracy landscapes (Fig. 4) and domain-wise gains (Fig. 5) are consistent with this picture, thereby strengthening the conceptual understanding even in the absence of full theory.
>
> Weakness 2 – Limited experimental analysis and lack of t-SNE plots
> Thank you for the great suggestion, we implemented a TNSE analysis and can see CONTXT causes same class samples to cluster better across contexts, highlighting the methods ability to move samples of differing contexts together. We will include these plots in the final version of the manuscript.
>
> Weakness 3 – Writing and paper structure
> Thank you very much for suggestions to improve the presentation of our results. Our intent in briefly re-describing CONTXT in the Results section was to provide a self-contained, intuitive reminder before diving into experiments, but we see how this can read as an abrupt reintroduction of the method.
> In the revision, we will:
> Remove redundant descriptions of the method from the Results section and replace them with concise pointers back to the Methods section.
> Tighten the exposition around line 180 and nearby paragraphs so that the transition from methods to experiments is smoother.
> Add a short “Intuition and limitations” subsection (either at the end of Methods or the beginning of Results) to centralize the conceptual discussion that is currently scattered.
>
> Overall
> We appreciate the reviewer’s assessment that the direction is promising. While our current focus is on empirical effectiveness and a clear operational recipe, we agree that deeper theory and additional analysis tools such as visualizations could further strengthen the work, and we will explicitly frame these as future extensions.

---

### Official Review · Reviewer_1oEf · 2025-10-31

**Soundness:** 2
**Presentation:** 2
**Contribution:** 2
**Rating:** 2
**Confidence:** 4

**Summary:**

This paper presents CONTXT, a method of activation steering that helps neural networks perform better in new domains. CONTXT first computes a context vector, then uses this context vector at inference time (by simple vector operations such as addition or subtraction) to modify the model's internal activations. The paper experiements on OOD image classification with VGG and text generation with LLMs.

**Strengths:**

* Simplicity: The method does not require re-training the model, but rather, steering activations at inference. This is a major selling point since model re-training/finetuning can be cumbersome and expensive.
* The core idea of the paper is presented well, and Figure 1 (while not exactly visually appealing) does a good job of highlighting the method clearly.

**Weaknesses:**

* The fundamental idea of activation steering is not entirely new. This concept has been applied in various different domains, including text generation and even other image generative models. The paper claims the novelty on the specific formulation and the application of their method to OOD classification. However, I am not convinced that CONTXT provides a fundamentally new concept, but rather an application of an existing concept.
* While the paper claims that there is little hyperparameter tuning, it still needs to be tuned per-layer, which is not trivial. The authors suggest that this could be learned in the validation setting. Furthermore, the claim that CONTXT requires "minimal computational overhead" or "minimal latency" is not in good faith, since computing context vectors themselves could be expensive depending on the setting. This leads me to my next point:
* CONTXT is motivated by Domain Generalization (DG) and also claims to experiment on DG settings; however, the experimental settings do not align with the core issues in DG. In DG, we assume that the target domain is not available at any time. Thus, since CONTXT requires data from a target domain to create the context vector, it cannot be classified as tacking the DG problem. It is closer to domain adaptation or test-time adaptation.
  * To add to the point above, the motivating example in Figure 2 is a bit misleading because it instills the idea that the "correct" context is known for the image. However, the correct context here depends entirely on the classification of the image ("cow on a beach"). So essentially, we need to have already classified the image to know which context to remove or add.

**Questions:**

I would appreciate authors' response to the weaknesses addressed.

---

> ### Author Response · Authors · 2025-11-27
>
> Thank you for the careful review and for highlighting both the strengths and limitations of our work. Below, we address each concern in turn.
>
> Weakness 1: Novelty of activation steering.
>
> We agree that activation steering as a general concept is not new, and we should have been more precise in the original text to avoid giving the impression that we introduced activation steering itself. Our intended contribution is a concrete, simple and still novel recipe (CONTXT) for doing it in a way that (i) requires no additional training, sparse autoencoders, or paired prompts, (ii) operates directly in feature space, and (iii) applies unchanged to both discriminative models (VGG-based classifiers) and generative LLMs.
> Prior steering work typically falls into one of two categories:
>
> Requires training auxiliary models (e.g., SAEs) or learning steering vectors; or
>
> Uses token-level differences between paired prompts, which need clean opposites and careful token alignment, and are primarily designed for LLMs.
>
> In contrast, CONTXT needs only forward passes to compute context vectors once, and then uses the same small set of vectors and scalars to steer both OOD image classification and LLM outputs. We will revise the paper to emphasize that our novelty is in this unified, training-free, context–vector formulation and its empirical demonstration across both settings, rather than in the existence of activation steering as a general idea.
>
>
> Weakness 2: Hyperparameters and computational overhead.
>
> CONTXT adds a scalar weight per index, but in our experiments we modify only one layer with at most two indices (“inject” in-domain, “remove” OOD), so there are only one or two scalar hyperparameters. A small grid search on a validation split is sufficient and far simpler than typical fine-tuning.
> Our “minimal overhead” claim refers to inference: context vectors require a one-time offline computation that can be cached, and deployment adds only a subtraction and a scaled addition per relevant layer. In practice, this cost is small relative to the model’s forward pass. We will clarify these points and soften the wording in the revision.
>
>
> Weakness 3: DG vs domain adaptation / test-time adaptation).
>
> We appreciate this point and agree that the strict DG formulation usually assumes no access to target-domain data. CONTXT, as instantiated in our experiments, does use unlabeled target-domain samples to build the “remove” context vector (and training-domain samples for the “inject” vector). As you note, this aligns more naturally with domain adaptation or test-time adaptation. We will update the terminology throughout to reflect this more accurately: we will retain DG as motivation (the performance problem we care about) but describe our setting as “adaptation under access to unlabeled target-domain examples” rather than as canonical DG.
>
> Response regarding Figure 2 and the “correct” context.
> The concern that the classification problem illustrated in Figure 2 presupposes knowledge of the “correct” context is understandable, and we will revise the figure and caption to avoid this confusion. In our formulation, class and context are distinct: the classification label for the example is an object label (“ox”/“cow”), whereas the contexts are environment domains such as “farm,” “beach,” or “city.” The intended setting is one where it is known from meta-data or domain knowledge that training images predominantly come from one environment (e.g., farms) and deployment images from another (e.g., beaches). Context vectors can then be computed from unlabeled domain-specific images (farm scenes, beach scenes), without knowing the true object label for the particular test image. While prior knowledge of the “correct” context may seem like a strong assumption, it is common to know the environment or domain of the training data.
>
> Thus, for the “cow on a beach” OOD image, CONTXT can improve accuracy by (globally) moving features toward the training environment (“farm”) and away from the deployment environment (“beach”) across the test set, not by conditioning on the true “cow” label of that specific image. The “city” control in Figure 2 illustrates that injecting an irrelevant context does not recover the correct class, reinforcing that we are not simply encoding the label. We will rewrite the text around Figure 2 and adjust the caption to emphasize that contexts are domain-level features derived from separate examples, not from the ground-truth class of the particular image.

---

### Official Review · Reviewer_8VGh · 2025-11-01

**Soundness:** 1
**Presentation:** 1
**Contribution:** 2
**Rating:** 2
**Confidence:** 4

**Summary:**

The paper proposes CONTXT, a lightweight “context steering” mechanism that edits intermediate activations by adding a scaled difference between a cached “context vector” and the current hidden state. The method aims to (a) improve domain generalization for vision classifiers and (b) steer LLM generations toward a desired attribute/persona without fine-tuning.

**Strengths:**

S1. Domain generalization using just a vector operation is a very solid research direction.

S2. The paper is generally clear apart from figures (refer weakness)

**Weaknesses:**

W1. "Brain-inspired" motivation is confusing: The biological framing (PFC top-down control) is used to justify adding a linear direction to features, but no concrete architectural, algorithmic, or intuitive notions are provided. I don't really understand how this is related to brain in any way.

W2. The construction of "context vectors" is largely a rehash of a very relevant line of literature on prototype-based/centroid-based learning, like [1] which is not discussed at all. Context vectors are similar to mean feature vectors which act as references and moves points toward/away from them via linear vectors - a clear similarity to using the similarity as the distance metric.

W3. "Context" needs to be better defined: The definition is changed across sections, making it hard to formalize what a valid context vector is. For language, it is defined as "The injected (in-domain) context vector comprised of the average feature representation across all training domain samples," - which does not make sense to me, is the entire training corpus a context? If this is true then why not do the same for images? Additionally, the paper gives formulas for single/multiple contexts, but not a rigorous selection/validation protocol for constructing them in general.

W4. The quality of figures is very poor, with no error bars, statistical tests, etc. reported.

[1] This Looks Like That: Deep Learning for Interpretable Image Recognition, NeurIPS 2019

**Questions:**

1. Can the authors explain in detail why their method is derived from brain functioning?

2. Were any other similar methods explored?

---

> ### Author Response · Authors · 2025-11-27
>
> We thank the reviewer constructive feedback, please see our responses below.
>
> W1: “Brain-inspired” motivation
> Our original description did not clearly explain the neuroscience link, which we now clarify. In the brain, seeing a “cow on the beach” produces ambiguous bottom-up input, so early visual areas extract features (four-legged body, gait, fur) that activate a broad “farm animal” representation in IT and anterior temporal cortex when a specific animal cannot be chosen. This high-level “farm” context, maintained in anterior temporal and prefrontal cortex , then sends top-down signals back to ventral visual areas, that enhance farm-related features and suppress irrelevant beach details, making recognition easier. CONTXT operates analogously: a contextual vector shifts the representation toward a “farm” manifold and away from a “beach” manifold, amplifying relevant features and suppressing misleading scene information so that the resulting representation is cleaner and easier to classify.
>
> W2: Relation to prototype / centroid methods
>
> We appreciate the pointer to [1] and agree that discussing prototype-based models will strengthen the paper; we will add an explicit comparison in the related-work section.
>
> However, we respectfully disagree that CONTXT is “largely a rehash” of that literature. Prototype methods such as ProtoPNet learn class- or part-specific prototypes (e.g., “cow horn,” “cow spots”), and classification is based on similarity to these learned prototypes; the prototypes are trained and tied to object identity. In contrast:
>
> In our DG experiments, each CONTXT vector is domain-level and label-agnostic: for PACS/CCT we average features over all classes within a domain (Photo / Art / Cartoon / Sketch, or different camera locations) and then apply the same context vector to every class. CONTXT is explicitly designed to capture contextual factors such as style, background, or environment rather than object parts.
>
> CONTXT never replaces the classifier with a prototype-similarity layer; instead, it linearly edits internal features of an existing network at inference time, leaving the architecture and decision rule unchanged.
>
> ProtoPNet and related interpretable-prototype architectures require training and have been developed for supervised image classification only, whereas CONTXT is training-free, applies to any model with accessible hidden states, and we show that the same mechanism steers both
> CNN classifiers and LLM generations.
>
> We will clarify these conceptual differences in the text.
>
>
> W3: Definition and construction of “context”
>
> We agree that the notion of “context” should be made more precise, and we will revise the text accordingly.
>
> In our framework, at a chosen layer \ell, a context \kappa is defined by a set of inputs that share some non-label attribute (domain, style, sentiment, persona, etc.). The context vector is then:
> c_{\ell,\kappa} \;=\; \frac{1}{|S_\kappa|}\sum_{x \in S_\kappa} h_\ell(x),
> where S_\kappa​ is the set of examples realizing context \kappa.
>
> Images (PACS/CCT). For each domain (e.g., Photo, Cartoon) we construct c_{\ell,\kappa}​ by averaging features over that domain, independent of class. At test time, if the model was trained on Photos but evaluated on Cartoons, we inject the Photo context and remove a Cartoon context computed from a disjoint validation split of the Cartoon domain.
>
> Language / LLMs. For LLMs, we similarly need examples that demonstrate a desired context. LLM feature representations are much more precise and averaging multiple examples results in unpredictable outputs so we use the hidden state of a short phrase that denotes the context (e.g., “Statue of Liberty”, “be extremely positive”) as a compact contextual proxy
>
> Importantly, the construction protocol is the same across modalities. We will make this unifying definition explicit and move the image/LLM instantiations into separate subsections to avoid apparent redefinitions.
>
>
> W4: Figure quality, error bars, and statistics
> We agree that the quantitative figures would be stronger with variance estimates. For the initial submission we reported single-seed runs due to compute and time constraints. For the final version, we will re-run experiments with multiple random seeds and report mean ± standard deviation.
>
>
>  Q1: Why is the method “derived from brain functioning”?
> Please, see W1.
>
>
> Q2: Were any other similar methods explored?
> Our work is closely related to activation-engineering / steering methods such as Activation. We discuss these in Section 1–3 and position CONTXT as a simpler that (i) does not require paired prompts or opposites, (ii) uses a single context vector per attribute, and (iii) is evaluated on both OOD classification and LLM style/persona control. We can elaborate this in the final revision.

---

### Meta-Review · Area_Chair_4Jyd · 2026-01-02

**Summary:**

The fundamental idea of activation steering is not entirely new and has been explored across multiple domains, including text generation and other image generative models. The paper positions its novelty in the specific formulation of the method and its application to out-of-distribution (OOD) classification. However, this contribution appears to primarily repurpose an existing concept rather than introduce a fundamentally new learning paradigm or theoretical insight. While the proposed CONTXT framework demonstrates reasonable effectiveness in the target task, the lack of conceptual novelty limits its significance, leading to the assessment that the work is better characterized as an application-driven extension of prior ideas rather than a substantive methodological advance, and therefore leans toward rejection.

**Reviewer Scores:**

No

---

### Decision · Program_Chairs · 2026-01-26

Reject